# Synthesis of RGO/Cu@ FeAl_2_O_4_ Composites and Its Applications in Electromagnetic Microwave Absorption Coatings

**DOI:** 10.3390/ma16020740

**Published:** 2023-01-12

**Authors:** Zhenhua Chu, Wenxing Deng, Jingxiang Xu, Fang Wang, Zheng Zhang, Qingsong Hu

**Affiliations:** Shanghai Engineering Research Center of Hadal Science and Technology, College of Engineering, Shanghai Ocean University, Shanghai 201306, China

**Keywords:** FeAl_2_O_4_, composite materials, microwave-absorbing materials, wide frequency, absorption mechanism

## Abstract

In order to satisfy the requirements of wide frequency bands, the lightweight and strong absorption for the electromagnetic wave absorbing materials, a uniform mixture of FeAl_2_O_4_ with RGO/Cu (reduction graphene oxide, RGO) was obtained by the mechanical mixing method, and composite coating was obtained by plasma spraying. The addition of RGO/Cu into FeAl_2_O_4_ is conducive to improve the dielectric properties and the impedance matching performance of spinel. When the RGO/Cu composite powders are doped by 10 wt.%, the reflection loss at 15 GHz is −16 dB and the absorption bandwidth is 2 GHz, indicating that the composite material has potential application value in the field of high-frequency wave absorption. The research on the electromagnetic wave absorption mechanism shows that its superior wave absorption performance is determined by the synergistic effect of multiple loss mechanisms such as interfacial polarization, dipole relaxation, natural resonance, exchange resonance, and eddy current loss.

## 1. Introduction

With the development of electronic equipment to satisfy the intelligent communication and smart devices, the electromagnetic frequency is becoming higher and higher. However, electromagnetic radiation has been reported to cause some damage to the human body. In the microwave frequency band absorbing materials perform large dielectric and magnetic losses which can be applied in absorption devices to decrease electronic contamination in the modern world [1,2,3,4]. Therefore, the development and application of electromagnetic-wave-absorbing composite materials to meet the requirements of wide frequency bands and lightweight and strong absorption are important [5,6,7,8].

Generally, in order to increase the attenuation and absorption of electromagnetic waves, materials with larger complex permittivity imaginary parts and complex permeability imaginary parts can be used [9,10,11]. Zhao et al. [12] used Al/Fe_2_O_3_ as feed spraying powders to prepare coatings with FeAl_2_O_4_ and Fe as the main phases by plasma spraying technology. The research shows that the increase in the conductivity of the coating is the main reason for the increase in the imaginary part of the complex permittivity. Trukhanov et al. [13,14,15] studied the electromagnetic properties of a series of ferrite-based materials and found that the magnetic property is influenced by the concentration of aluminum ions in BaFe_12−x_Al_x_O_19_ (x = 0.1–1.2) [16]. Its good microwave-absorbing property results from the good conductivity.

However, high density is also an important issue which limits their application. As a light material with excellent conductive carbon materials, reduced graphene oxide (RGO) has attracted extensive attention for new electromagnetic wave absorption [17]. Gu et al. [18] reported that single crystalline hollow Fe_3_O_4_ spheres were grown onto RGO flakes with remarkable microwave absorption properties. Xu et al. [19] obtained good microwave absorption for PPy/rGO-0.6, and the Co_15_Fe_85_@C/RGO composite showed an excellent microwave absorption performance [20].

However, good conductivity materials do not effectively satisfy the impedance matching mechanism. Therefore, in order to meet the strong absorption and broad band requirement of the absorbing materials, composite materials are combined with magnetic loss, dielectric loss, and conductivity to further adjust the electromagnetic parameters to broaden the absorbing band. The FeAl_2_O_4_ is an electromagnetic-wave-absorbing material due to its double-complex dielectric properties with dielectric loss and magnetic loss capabilities. It can be adopted as a carrier of composites to adjust the conductivity and magnetic permeability.

On the other hand, the structure of composites are favorable to enhance the stability of absorbers [21,22,23,24]. Reduced graphene oxide (RGO) has great utility in the field of electromagnetic wave absorption due to its residual defects and functional groups [25]. Shu et al. [26] prepared magnesium ferrite microspheres modified nitrogen-doped reduced graphene oxide hybrid composites by solvothermal and hydrothermal methods.

The combination of ferrite and graphene to achieve a new lightweight, broad band electromagnetic-wave-absorbing material has become a new trend. In the present study, composite coatings were combined with magnetic loss and dielectric loss FeAl_2_O_4_, non-magnetic reduced graphene oxide and good conductivity Cu. The electromagnetic parameters of composites were adjusted by controlling the proportion of three components. On the other hand, the microwave absorption frequency of the absorbing material was enlarged by the structure of the composites. In the gigahertz frequency range, the introduction of RGO/Cu high dielectric constant materials was helpful to enhance the dielectric loss of spinel ferrites. Graphene obtained by the thermal reduction of graphene oxide not only improved the impedance matching performance of the material, but also produced defect polarization relaxation and electromagnetic dipolar polarization relaxation of oxygen-containing functional groups, which led to the improvement of the wave-absorbing effect of the composite coating. Therefore, the synthesized hybrid structure is considered to be a potential absorbing material.

## 2. Experimental Process

Electrofused FeAl_2_O_4_ spinel was purchased from Henglong Co., Ltd., Guangzhou, China. Because the size of the electric smelting FeAl_2_O_4_ spinel is large for spraying, the planetary ball mill was adopted to reduce to the size of the smelting FeAl_2_O_4_ spinel ferrite, and then the particles with a size of 10–30 µM were sieved. Then, spray granulation drying technology was adopted to obtain the Cu/GO composite powders. The commercially available copper powders and graphene oxide powders were weighed according to the mass ratio M_Cu_: M_GO_ = 9:1. During the spray granulation process, the solid content of the slurry was 35 wt.%, the temperature of the drying chamber was 230 °C, and the speed of the peristaltic pump was 25 r/min. Compared with graphene, graphene oxide contains a large number of oxygen-functional groups, so that it has superior dispersibility in water, which is conducive to the uniformity and stability of the slurry in the gas-automizing process. After that, the powders were placed in an argon-protected vacuum tube furnace for low-temperature thermal reduction. The vacuum tube furnace was charged with argon for thermal reduction. The heating rate was 5 °C/min, the target temperature was 350 °C, the temperature was kept for 1 h, the cooling rate was 5 °C/min, and the reduction was completed after the temperature was lowered to room temperature. Finally, the RGO/Cu composite powders produced after reduction was mechanically mixed with FeAl_2_O_4_, and the mixing ratios were 5%, 10%, and 15%, respectively. The process of the fabrication of composite powders is shown in Figure 1. Then, the coating was fabricated by the plasma spraying process. On this basis, a control group was added with the same Cu content as the first group. The comparison group was compared with the first group, and RGO was used as the only variable to further study the influence of RGO on the absorbing performance. In this paper, samples are named as B1, B2, B3, B4, and B5, respectively, and the contents of samples are summarized in Table 1.

The morphologies of the samples were observed by field emission scanning electron microscopy ((SEM, S4800, Hitachi, Tokyo, Japan)). Different phases of composite powders were identified by X-ray diffraction (XRD) with Cu-Ka radiation (XRD, Bruker D8 Focus, Billerica, MA, USA). The reduction degree of Cu/GO was characterized by Raman spectroscopy (Thermo Fischer DXR, Bedford, MA, USA) and Fourier transform infrared spectroscopy (Thermo Scientific Nicolet IS5, Bedford, MA, USA). The laser wavelength of Raman spectroscopy was 633 nm. The magnetic properties were investigated by a vibrating sample magnetometer (VSM, Lakeshore 7307 Westerville, OH, USA) using the coaxial method. The electromagnetic parameters were measured by a vector network analyzer (Agilent N5222A, Medford, MA, USA). The details are as follows: FeAl_2_O_4_ and RGO/Cu composite powders were mixed with paraffin wax by the amounts of 50 wt.%. The final compounds were compacted into toroidal shapes with an inner diameter of 3.04 mm, an outer diameter of 7.0 mm, and a thickness of 2 mm.

## 3. Results and Discussions

### 3.1. Morphologies and Microstructure

The surface morphologies of the FeAl_2_O_4_ powders after ball milling are shown in Figure 2a, which are mostly spherical and regular in shape. The particle-size distribution of the powders is approximately 10–30 μM by sieving. Figure 2b shows the SEM micrograph of the RGO/Cu composite powders. A layer of film-like graphene oxide can be seen clearly around the copper particles (as marked in Figure 2b). The average particle size of the composite powders is approximately 10–15 μM in diameter. Each feedstock particle consists of many micro-sized Cu and GO granules and the composite particles are compact. Such structure of the composite powders provides homogeneous distribution of Cu and GO.

In the present study, a low-temperature solid-phase thermal reduction method is adopted to reduce Cu/GO powders. Excessive reduction temperature causes graphene to agglomerate and produces wrinkles, destroying the layer structure of graphene [15]. Figure 3a shows the Fourier infrared spectrum before and after the reduction of Cu/GO. This spectrum can directly reflect the types of groups contained in graphene oxide. It can be seen from the figure that graphene oxide has obvious absorption peaks at 3440 cm^−1^, 1630 cm^−1^, 1400 cm^−1^, and 1085 cm^−1^, which correspond to -OH, C=C, C-OH, and C-O-C chemical bonds. It indicates that graphene oxide has a large number of oxygen-containing groups. After thermal reduction at the argon atmosphere, the content of each oxygen-containing group is reduced. Combined with the Raman spectrum shown in Figure 3b, the area ratio ID/IG of peak D to peak G decreased from 1.79 to 1.22 after the reduction. The D band corresponds to the disorder induced by carbon atoms or the defects in the lattice structure. The G band can be attributed to the in-plane tensile vibration of the sp^2^ hybridization of carbon atoms, which represents the degree to which the material approximates the graphite structure. It is worthy to note that the area intensity (ID/IG value) of the D and G bands is related to the defects of the carbon atom crystal [27,28]. The higher the ID/IG value, the higher the degree of defects and the lower the degree of graphitization. The decrease in ID/IG value indicates that a new sp^2^ domain is produced during the reduction process, and the carbon atom lattice defects are reduced, which is beneficial to improve impedance matching and generate more polarization centers to attenuate electromagnetic waves. Combining the results of Fourier infrared spectroscopy and Raman spectroscopy, it can be seen that the Cu/GO composite powders have been reduced to a certain extent.

Figure 4 is the X-ray diffraction patterns of RGO/Cu-mixed FeAl_2_O_4_ as coatings after spraying on the steel substrate, which is mainly FeAl_2_O_4_ and Cu. Meanwhile, with the proportion increase in the RGO/Cu composite powders, the diffraction peak of copper gradually increases. Generally, RGO peaks cannot be detected by XRD. It usually is represented by the binding component [29]. On the other hand, the diffraction peak of copper oxide was not detected. This indicates that copper powders are not oxidized during granulation and thermal reduction. At the same time, there was no diffraction peak at 2θ = 26° and 2θ = 10°, indicating that there was no large-scale agglomeration of reduced graphene oxide during argon reduction, and its monolayer or few-layer structure was still maintained.

### 3.2. Magnetic Properties

The hysteresis loop is the closed magnetization curve of the magnetic hysteresis phenomenon when the magnetic field intensity changes periodically. It is an important basis for judging the magnetic properties of materials. Figure 5 shows the sample magnetization curve spectrum measured by a vibrating sample magnetometer (VSM) at a temperature of 300 K. The saturation magnetization (Ms), remanence magnetization (Mr), and coercive force (Hc) are listed in Table 2. It can be observed that all the samples presented a rather narrow hysteresis loop, suggesting a behavioral characteristic of soft magnetic features and all the samples have low saturation magnetization (Ms), which may be induced by their larger particle sizes [30,31]. With the increase of RGO/Cu ratio, the value of Ms decreased from 6.33 emu/g to 5.47 emu/g, mainly due to the increase in the proportion of copper powders with excellent electrical conductivity and the relative decrease in the amount of ferromagnetic FeAl_2_O_4_ ferrite.

### 3.3. Microwave Absorption Properties

In general, the complex permittivity (ε_r_ = ε′ − jε″) and complex permeability (μ_r =_ μ′ − jμ″) of the material determine the absorbing properties. The real part of complex permittivity and permeability (ε′ and μ′) represents the ability to store charge or energy, while the imaginary part (ε″ and μ″) represents the energy loss caused by the change of dipole moment or magnetic moment [24,32]. The dielectric and magnetic dissipation factors tanδ_e_ and tanδ_m_ were calculated by the ratio of imaginary part to the real part of permittivity and permeability, respectively [33,34]. Figure 6 shows the frequency dependence of the complex permittivity and complex permeability of the samples between 1 GHz and 18 GHz. Meanwhile, comparing the sample, in the data for a measurement cell without sample material only paraffin wax is obtained, as shown in Figure 6. The results show that the ε′ of the composite powders increase while the μ′ slightly decrease with the introduction of RGO/Cu composite powders. Combined with SEM and XRD, the corrugated structure of RGO can protect micron-sized copper from excessive oxidation, which is conducive to the formation of stable RGO/Cu composites. At the same time, the synergy between RGO and Cu is beneficial to enhance the electrical conductivity of the material and has a stronger ability to attenuate electromagnetic waves [34]. Compared with B1, the increase ε′ of the composites can be attributed to the improved electrical conductivity of the RGO/Cu. The values of ε″ for B4 first increase to a maximum at 15 GHz, followed by a continued decline with significant fluctuations (Figure 6b). This is attributed to the dielectric polarization and relaxation in the higher frequency range. Meanwhile, the imaginary part of the complex permeability of the B4 sample has an obvious peak around 15 GHz, as shown in Figure 5e, which is caused by eddy current loss and natural resonance in a higher frequency range.

It is worthy to note the dielectric loss and magnetic loss, when evaluating the microwave attenuation loss. The dielectric loss tangent and the magnetic loss tangent are shown in Figure 6c,f. The dielectric loss tangent curve and magnetic loss curve of all samples show a fluctuating state, and the dielectric loss tangent curve and magnetic loss curve of the B4 sample have a maximum value at 15 GHz. The dielectric loss tangent fluctuates in the range of 0 to 0.14, and the magnetic loss tangent fluctuates in the range of 0 to 0.45, indicating that the attenuation of electromagnetic waves is dominated by magnetic loss.

To further understand the effect of magnetic loss on microwave absorption performance, the following equation is cited [35]:(1)$$C_{0}=\mu^{\prime \prime }{\left(\mu^{\prime }\right)}^{-2}f^{-1}=2\pi \mu_{0}\sigma d^{2}/3$$
where *μ*_0_ is the magnetic permeability in vacuum, *σ* is the electrical conductivity of the material, and d is the thickness of the material. If the magnetic loss of the absorbing material is only caused by eddy current loss, then *C*_0_ should be constant in different frequency ranges [1,36].

As shown in Figure 7, the eddy current loss of each sample decreases with the frequency in the range of 1–8 GHz, and the eddy current loss is not the main loss mechanism. In the range of 8–18 GHz, except for the B4 sample, the loss curves of other samples remain constant, indicating that the eddy current loss is the main loss mechanism in this range. However, the magnetic loss of B4 sample should be ascribed to the exchange resonance. The imaginary part of the sample’s complex permeability (μ″) shows multiple resonance peaks. This may be due to the wide particle-size distribution of the composite powders, which results in different resonance frequency positions.

The dielectric loss ability mainly stems from electrical conductivity loss and polarization relaxation loss. Ion polarization and electron polarization usually occur in the frequency range of ultraviolet, visible, and infrared light. It can be ignored in the frequency range of microwave. Dipoles relaxation polarization refers to the polarization caused by the rotation of the dipole moment in the direction of the electric field, and it can greatly influence the dielectric loss. The relaxation loss can be analyzed by the Debye equation. Based on the Debye theory, the relationship between ε′ and ε″ can be plotted according to the equation:(2)$${\left[\varepsilon^{\prime }-\left(\varepsilon_{s}+\varepsilon_{\infty }\right)/2\right]}^{2}+{\left(\varepsilon^{\prime \prime }\right)}^{2}={\left[\left(\varepsilon_{s}-\varepsilon_{\infty }\right)/2\right]}^{2}\;$$
where *ε_s_* is the static permittivity, *ε_∞_* is the relative dielectric permittivity at the high-frequency limit, and each Cole–Cole semicircle corresponds to one Debye relaxation process [9,35,36].

As shown in Figure 8, with the RGO/Cu ratio gradual increase, the number of Cole-Cole semicircles decreases. When the doping ratio of RGO/Cu composite powders is 10 wt.%, an obvious semicircle appears, and the number of semicircles is the least among the samples. In addition, there are many irregular semicircles in various samples, indicating that there are other loss mechanisms in the absorption process.

To evaluate the microwave absorption performance, the as-obtained complex permittivity and permeability were used to determine the reflection loss (RL), based on the model of a single-layered plane wave absorber as proposed by Naito and Suetake [35,36]:(3)$$RL\left(dB\right)=20lg\left|\left(Z_{in}-Z_{0}\right)/\left(Z_{in}+Z_{0}\right)\right|\;$$
(4) Zin=Z0(μr/εr)tanh[j(2πfdc)(μrεr)]
where *Z_in_* is the normalized input impedance of a microwave absorber, *Z*_0_ is the characteristic impedance of free space (376.7 Ω), *ε*_0_ is the dielectric constant of free space, *μ*_0_ is the permeability of free space, and *c* is the velocity of light in free space (3 × 10^8^ m s^−1^).

According to Equations (3) and (4), the dip in RL is equivalent to the occurrence of minimum reflection or maximum absorption of the microwave power for a thickness. To investigate the influence of the addition of RGO/Cu on the electromagnetic wave absorption performance of the iron–aluminum spinel, the reflection loss curve of the B1–B5 samples at a filler content of 50 wt.% was studied. As presented in Figure 9, the RL_min_ values of B1, B2, B3, B4, and B5 are −4.2, −3.75, −3.45, −16.0, and −3.5 dB, respectively. With the ratio of Cu/RGO composite powders increase, the reflection loss of the material shows a trend of decreasing, firstly, then increasing, and decreasing finally. When the RGO/Cu doping ratio is 10 wt.% and the thickness is 3 mm, the electromagnetic wave absorption capacity is the best. The reflection loss reaches the minimum value of −15 dB at 15 GHz, and the effective bandwidth is 2 GHz. The reason for the large reflection loss of the B5 sample can be attributed to the excessive doping ratio of RGO/Cu composite powders to form a conductive grid, which generates a macro current on the surface of the material, causing the skin effect to prevent the incidence of electromagnetic waves [37]. For an electromagnetic wave absorbing materials, first of all, it should be made sure that electromagnetic waves completely enter the absorber, and then the electromagnetic energy of the incident electromagnetic wave is completely converted into heat energy in the absorber, through a series of loss mechanisms such as induced conduction current loss, dielectric loss, and magnetic loss. Therefore, the design of an absorbing electromagnetic-wave-absorbing coating must consider two aspects: impedance matching conditions and attenuation conditions. Impedance matching and attenuation conditions are described as the Equations (5) and (6) [38]:(5)Z=|Zin/Z0|=(μr/εr)tanh{j(2πfdc)(μrεr)}
(6)$$\alpha =\left(\sqrt{2}\pi f\right)/c\sqrt{(\mu^{\prime \prime }\varepsilon^{\prime \prime }-\mu^{\prime }\varepsilon^{\prime })+\sqrt{{\left(\mu^{\prime \prime }\varepsilon^{\prime \prime }-\mu^{\prime }\varepsilon^{\prime }\right)}^{2}+{\left(\mu^{\prime \prime }\varepsilon^{\prime \prime }+\mu^{\prime }\varepsilon^{\prime }\right)}^{2}}}$$

As shown in Figure 10a, for a single iron–aluminum spinel absorbing material, the impedance matching increases as the frequency increases. When the frequency is higher than 14 GHz, the impedance matching value is much greater than the best impedance matching value 1.0. In the case of the same copper powder doping ratio, the impedance matching value of the B2 sample is greater than that of the B3 sample in the range of 2–18 GHz, indicating that the introduction of reduced graphene oxide is beneficial to reduce the reflection of electromagnetic waves to a certain extent. With the increase in the doping ratio of RGO/Cu composite powders, the impedance matching value shows a trend of first decreasing and then increasing. In the range of 14–18 GHz, the impedance matching value of the doped 10 wt.% RGO/Cu composite powders is lower than the other samples, indicating that the addition of RGO/Cu powders can improve the impedance matching problem of a single spinel in the high-frequency range. The increase in the slope of the impedance matching value of the B5 sample can be attributed to the increase in the reflection of electromagnetic waves due to the skin effect. As shown in Figure 10b, the attenuation curves of the samples all present a fluctuating state. In the range of 0–12 GHz, the changing state of the samples is the same; when the frequency is higher than 12 GHz, the B4 sample has three obvious peaks, and the electromagnetic wave attenuation ability is the best.

Based on the above analysis, the possible attenuation mechanisms of EM waves for FeAl_2_O_4_ ferrite doped with 10 wt.% RGO/Cu composite powders are described in Figure 11. The superior absorption performance can be determined by the synergetic effect of various factors [2]. Firstly, a lower impedance matching value is conducive to electromagnetic waves entering the absorber, and the complex surface of the RGO/Cu composite powders can provide more reflection and scattering. Secondly, FeAl_2_O_4_ ferrite with double complex characteristics evenly dispersed in paraffin can induce natural resonance and exchange resonance. The defects and residual oxygen-containing groups in RGO can not only improve the impedance matching performance of the material but also produce defect polarization relaxation. Electromagnetic dipole polarization relaxation with oxygen-containing functional groups effectively improves the wave-absorbing performance of the material. Moreover, the interfaces such as RGO/Cu, FeAl_2_O_4_/RGO/Cu, FeAl_2_O_4_/Paraffin, and RGO/Cu/Paraffin could induce abundant interfacial polarization [39]. Based on the above advantages, FeAl_2_O_4_/Cu/RGO composite material is expected to become an ideal electromagnetic-wave-absorbing material.

## 4. Conclusions

In this paper, RGO/Cu composite powders with complex surface were successfully prepared by spray granulation and thermal reduction. The electromagnetic parameters, impedance matching, attenuation capability, and microwave absorption performance are highly dependent on the chemical compositions. Some conclusions are summarized as following:
The best absorption performance was obtained for the FeAl_2_O_4_ composite doped with 10 wt.% RGO/Cu composite powder. The reflection loss of the composite at 15 GHz can reach −16 dB and the effective bandwidth is 2 GHz.The effect of the introduction of RGO/Cu is an increase in the conductivity of the composite material due to its good conductivity. On the other hand, the complex surface helps to increase the reflection and scattering of electromagnetic waves in the absorption body. In addition, the introduction of RGO/Cu helps to increase the heterogeneous interface polarization, defect polarization, and conductivity loss of the composites.The results show that the introduction of RGO/Cu composite powder can effectively optimize the impedance matching of single FeAl_2_O_4_ ferrite and improve the electromagnetic wave absorption performance.


## Figures and Tables

**Figure 1 materials-16-00740-f001:**
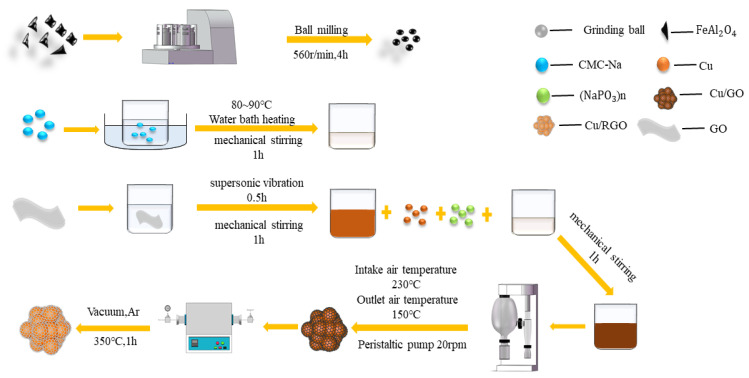
The description of the composite powder preparation process.

**Figure 2 materials-16-00740-f002:**
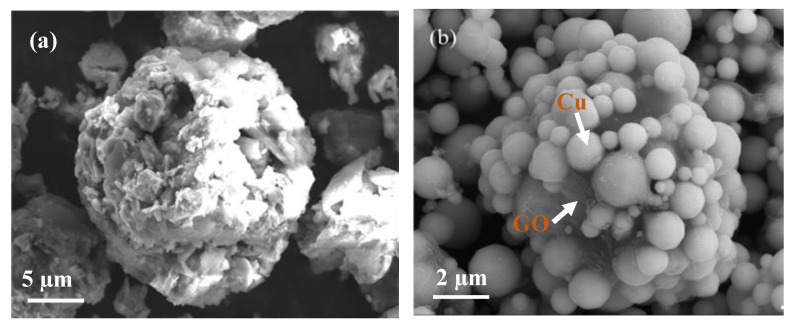
SEM morphologies of (**a**) FeAl_2_O_4_ powders and (**b**) Cu/GO composite powders.

**Figure 3 materials-16-00740-f003:**
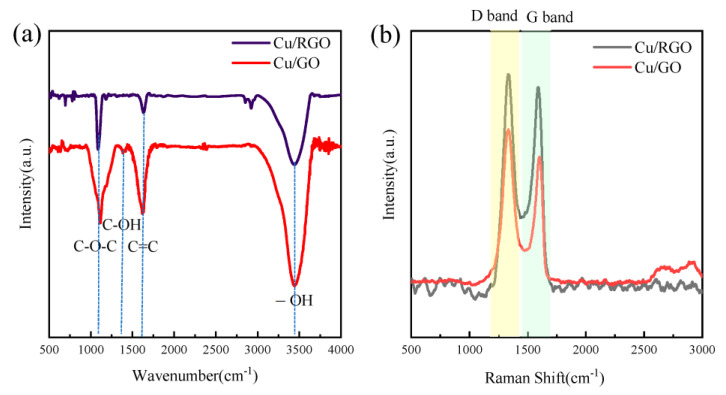
Fourier infrared spectrum (**a**) and Raman spectrum of the sample (**b**) of Cu/GO and Cu/RGO composite powders.

**Figure 4 materials-16-00740-f004:**
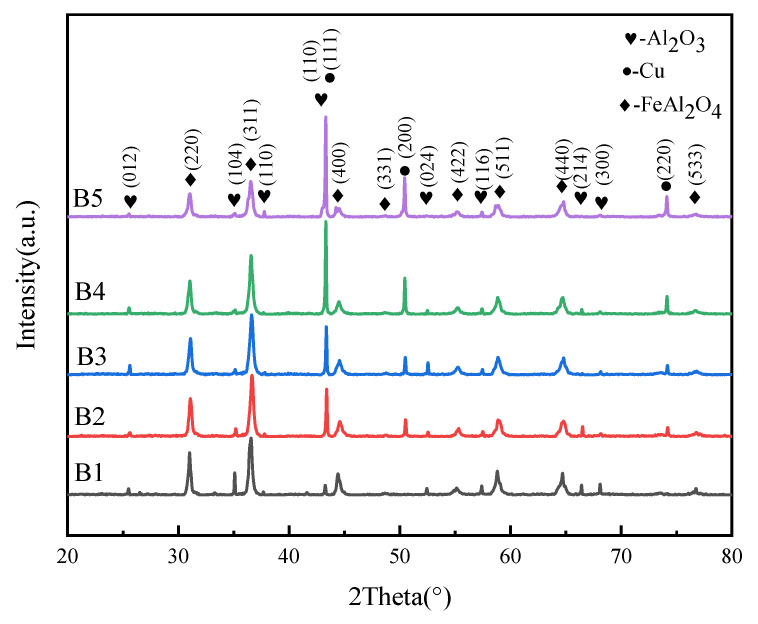
XRD patterns of composite coatings.

**Figure 5 materials-16-00740-f005:**
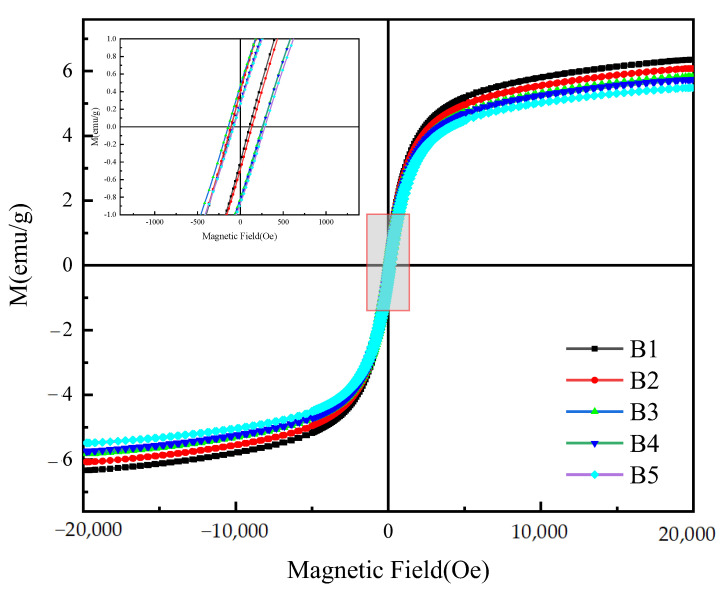
Hysteresis loops of the composite powders with different contents and the enlarged view as shown in the inset.

**Figure 6 materials-16-00740-f006:**
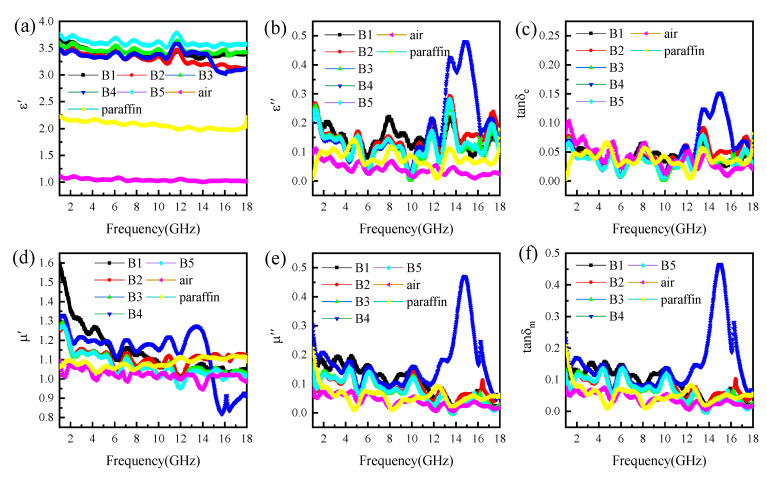
Frequency dependence of complex relative permittivity (**a**–**c**) and permeability (**d**–**f**) for the paraffin composites at a fixed absorbent content of 50 wt.%.

**Figure 7 materials-16-00740-f007:**
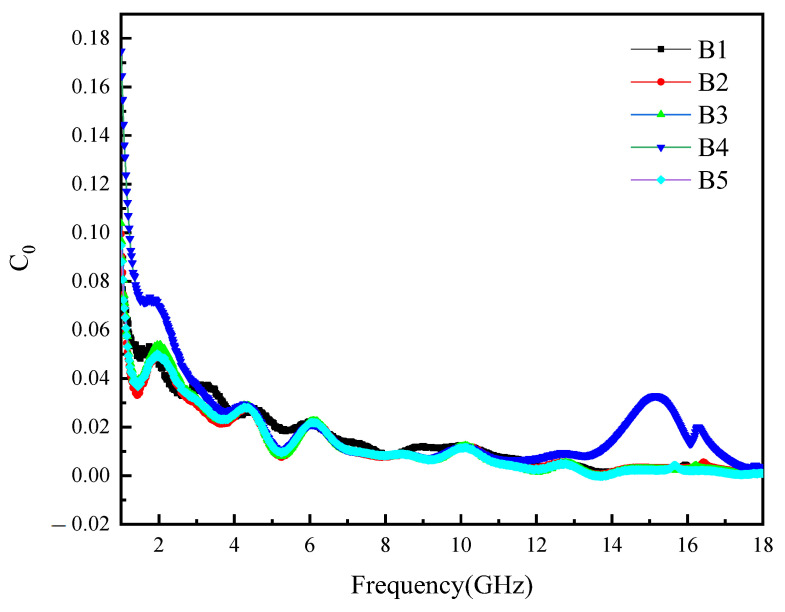
Eddy current loss of the samples with different contents.

**Figure 8 materials-16-00740-f008:**
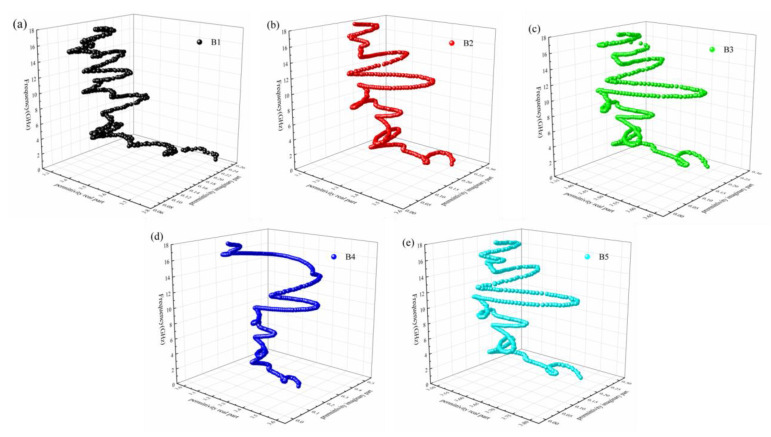
The Cole-Cole curves of paraffin composites B1 (**a**), B2 (**b**), B3 (**c**), B4 (**d**), and B5 (**e**).

**Figure 9 materials-16-00740-f009:**
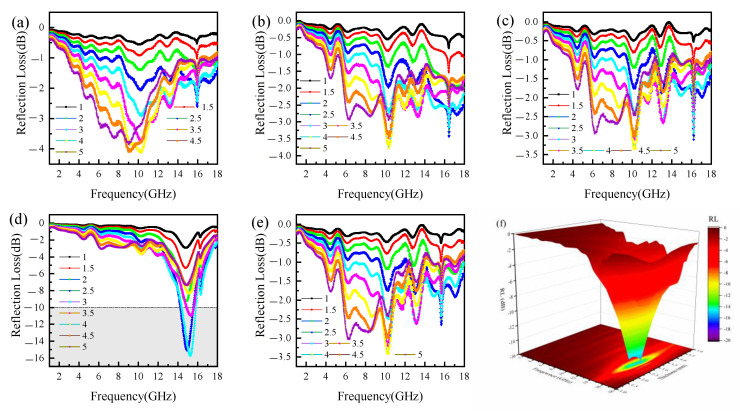
Reflection loss curves of the samples at different thicknesses: (**a**) B1, (**b**) B2, (**c**) B3, (**d**) B4, (**e**) B5, and (**f**) three-dimensional representation of reflection loss of B4.

**Figure 10 materials-16-00740-f010:**
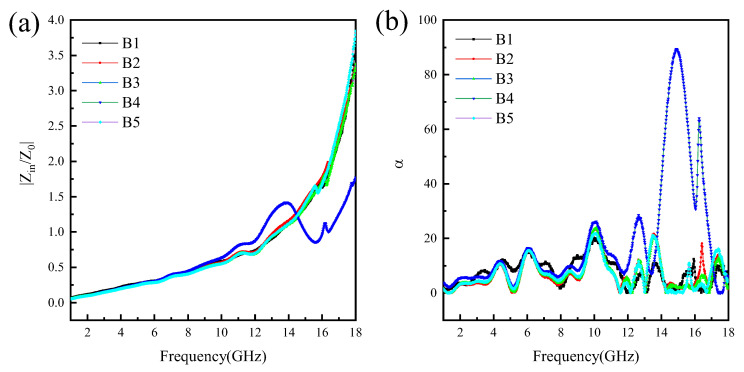
Impedance matching profiles (**a**) and attenuation constants (**b**) of samples.

**Figure 11 materials-16-00740-f011:**
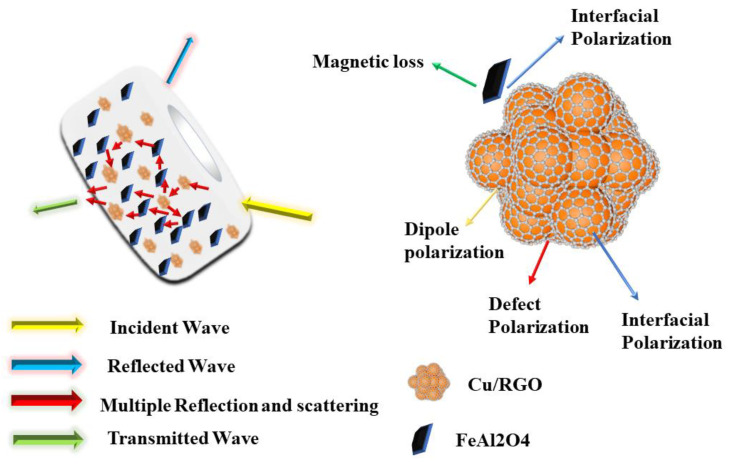
Schematic of microwave absorption mechanisms of the FeAl_2_O_4_/Cu/RGO.

**Table 1 materials-16-00740-t001:** The summarized content of samples.

	B1	B2	B3	B4	B5
FeAl_2_O_4_ (wt.%)	100	95.5	95	90	85
RGO/Cu (wt.%)	0	0	5	10	15
Cu (wt.%)	0	4.5	0	0	0

**Table 2 materials-16-00740-t002:** Magnetic parameters of the composite powder samples.

	Ms (emu/g)	Mr (emu/g)	Hc
B1	6.33	0.40	106.88
B2	6.08	0.45	121.70
B3	5.83	0.65	200.60
B4	5.74	0.56	174.02
B5	5.47	0.56	182.73

## Data Availability

Not applicable.

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
