# Peer review of "Synthesis of RGO/Cu@ FeAl2O4 Composites and Its Applications in Electromagnetic Microwave Absorption Coatings"

_materials, 2023, doi:10.3390/ma16020740_

Round 1
Reviewer 1 Report
The authors have written a research article entitled “Synthesis of RGO/Cu@ FeAl2O4 Composites and its Applications in Electromagnetic Microwave Absorption Coatings”. The manuscript is quite interesting, well framed, and based on the mixture of FeAl2O4 ferrite with RGO/Cu (Reduction Graphene Oxide, RGO) was obtained by the mechanical mixing method and composite coating was obtained by plasma spraying. Furthermore, the electro-magnetic wave absorption mechanism exhibits that its superior wave absorption performance is determined by the synergistic effect of multiple loss mechanisms such as interfacial polarization, di-pole relaxation, natural resonance, exchange resonance and eddy current loss. The work reported in this manuscript is very interesting and very well-presented. The authors have described the concept to a greater extent but the manuscript still needs few Minor corrections before acceptance in Materials.
I appreciate the author's effort in this good study. However, the following minor comments need to be addressed.
Comment 1: Grammatical/typographical error issues are so many there in the manuscript at several places also check superscripts and subscripts errors. For e.g. ine 56, “FeAl2O4” should be “FeAl2O4”
Comment 2: The introduction provided a good, generalized background of the topic that quickly gives the reader an appreciation. However, I think the authors need to cite and dscuss the below mentioned references in the introduction section to strengthen the section.
Bao, Susu, Wen Tang, Zhijia Song, Qiaorong Jiang, Zhiyuan Jiang, and Zhaoxiong Xie. "Synthesis of sandwich-like Co 15 Fe 85@ C/RGO multicomponent composites with tunable electromagnetic parameters and microwave absorption performance." Nanoscale 12, no. 36 (2020): 18790-18799.
Fabrication of PPy Nanosphere/rGO Composites via a Facile Self-Assembly Strategy for Durable Microwave Absorption.
Cui, Guangzhen, Yanli Lu, Wei Zhou, Xuliang Lv, Jiangnan Hu, Guoyu Zhang, and Guangxin Gu. "Excellent microwave absorption properties derived from the synthesis of hollow Fe3O4@ reduced graphite oxide (RGO) nanocomposites." Nanomaterials 9, no. 2 (2019): 141.
Comment 3: Include the peak positions in the IR spectrum (Figure 3a).
Comment4: In section 3.1., compare and discuss the XRD analysis results with relevant studies.
Comment 5: To confirm the elemental composition and bonding configurations perform the XPS spectra of as-synthesized RGO/Cu@FeAl2O4 composites.
Comment 6: Please improve the conclusion section.
Comment 7: The homogeneity of the references section needs to be maintained. So please check and revise accordingly to the journal's instructions.
Author Response
Comment 1: Grammatical/typographical error issues are so many there in the manuscript at several places also check superscripts and subscripts errors. For e.g. ine 56, “FeAl2O4” should be “FeAl2O4”
- As pointed by the reviewer, all of them have been corrected.
Comment 2: The introduction provided a good, generalized background of the topic that quickly gives the reader an appreciation. However, I think the authors need to cite and dscuss the below mentioned references in the introduction section to strengthen the section.
Bao, Susu, Wen Tang, Zhijia Song, Qiaorong Jiang, Zhiyuan Jiang, and Zhaoxiong Xie. "Synthesis of sandwich-like Co15Fe85@C/RGO multicomponent composites with tunable electromagnetic parameters and microwave absorption performance." Nanoscale 12, no. 36 (2020): 18790-18799.
Fabrication of PPy Nanosphere/rGO Composites via a Facile Self-Assembly Strategy for Durable Microwave Absorption.
Cui, Guangzhen, Yanli Lu, Wei Zhou, Xuliang Lv, Jiangnan Hu, Guoyu Zhang, and Guangxin Gu. "Excellent microwave absorption properties derived from the synthesis of hollow Fe3O4@ reduced graphite oxide (RGO) nanocomposites." Nanomaterials 9, no. 2 (2019): 141.
- As suggested by the reviewer, all the references have been discussed in the Introduction part.
Comment 3: Include the peak positions in the IR spectrum (Figure 3a).
- As suggested by the reviewer, the peak positions have been modified in the revised manuscript. And the values of the peaks also were presented in the text.
Comment4: In section 3.1., compare and discuss the XRD analysis results with relevant studies.
- As suggested by the reviewer, the discussion about XRD results have been added into the revised manuscript.
Comment 5: To confirm the elemental composition and bonding configurations perform the XPS spectra of as-synthesized RGO/Cu@FeAl2O4 composites.
- Yes. As suggested by the reviewer, the XPS is an important method to confirm the elemental composition, especially for the bonding configurations. However, at present situation, we can’t get good results of XPS. We are going to study the samples by XPS in our following work. On the other hand, we think the results of Fourier infrared spectrum and Raman spectrum also can offer the bonding configurations in the composites.
Comment 6: Please improve the conclusion section.
- As suggested by the reviewer, the conclusion section has been modified.
Comment 7: The homogeneity of the references section needs to be maintained. So please check and revise accordingly to the journal's instructions.
- As pointed by the reviewer, all the references have been corrected.
Reviewer 2 Report
The article "Synthesis of RGO/Cu@FeAl2O4 composites and its application in coatings that absorb electromagnetic microwave radiation" presents studies on the synthesis and use of composites for the absorption of microwave radiation. This work contains detailed data that reflects a certain amount of work. Some important experimental data and conclusions require verification and correction.
1. The authors claim that FeAl2O4 is a ferrite, but the general formula of ferrite is MFe2O4. This needs to be corrected.
2. Many research papers have reported the use of spinels. It is necessary to emphasize the novelty of the work and point out the improvement compared to the reporting work. For example DOI 10.1149/2162-8777/acaeb8.
3. The authors state that “The FeAl2O4 composite doped with 331 10 wt.% RGO/Cu composite powder has the best absorption effect. The reflection loss of the composite at 15 GHz can reach -16 dB and the effective bandwidth is 2 GHz”. However, the coating thickness is 5 mm, i.e. -3.1 dB/mm, which is not significant in comparison with analogues.
Author Response
- The authors claim that FeAl2O4 is a ferrite, but the general formula of ferrite is MFe2O4. This needs to be corrected.
- As pointed by the reviewer, we have corrected the formula.
- Many research papers have reported the use of spinels. It is necessary to emphasize the novelty of the work and point out the improvement compared to the reporting work. For example DOI 10.1149/2162-8777/acaeb8.
- As suggested by the reviewer, the reference is excellent work. We have read it and cited it.
- The authors state that “The FeAl2O4 composite doped with 331 10 wt.% RGO/Cu composite powder has the best absorption effect. The reflection loss of the composite at 15 GHz can reach -16 dB and the effective bandwidth is 2 GHz”. However, the coating thickness is 5 mm, i.e. -3.1 dB/mm, which is not significant in comparison with analogue
- As marked in our manuscript, the thickness of test sample is 2 mm “The final compounds were compacted into toroidal shapes with an inner diameter of 3.04 mm, an outer diameter of 7.0 mm, a thickness of 2 mm. “